# *Moringa oleifera* Extract Extenuates *Echis ocellatus* Venom-Induced Toxicities, Histopathological Impairments and Inflammation via Enhancement of Nrf2 Expression in Rats

**Akindele O. Adeyi** [1,*], **Sodiq O. Adeyemi** [1], **Enoh-Obong P. Effiong** [1], **Babafemi S. Ajisebiola** [2], **Olubisi E. Adeyi** [3] **and Adewale S. James** [3]

1   Animal Physiology Unit, Department of Zoology, University of Ibadan, Ibadan P.M.B. 200284, Oyo State, Nigeria; yemity09@gmail.com (S.O.A.); Enop@yahoo.com (E.-O.P.E.)
2   Department of Zoology, Osun State University, Oshogbo P.M.B. 230212, Osun State, Nigeria; dbabslinks@yahoo.com
3   Department of Biochemistry, Federal University of Agriculture, Abeokuta P.M.B. 2240, Ogun State, Nigeria; bisi_resourceful@yahoo.com (O.E.A.); Whljaymz@gmail.com (A.S.J.)
*   Correspondence: delegenius@yahoo.com; Tel.: +234-80-3069-2698

**Abstract:** *Echis ocellatus* snakebite causes more fatalities than all other African snake species combined. *Moringa oleifera* reportedly possesses an antivenom property. Therefore, we evaluated the effectiveness of *M. oleifera* ethanol extract (MOE) against *E. ocellatus venom* (EOV) toxicities. Thirty male rats were grouped as follows (*n* = 5): Group 1 (normal control received saline), groups 2 to 6 were administered intraperitoneally, 0.22 mg/kg (LD50) of EOV. Group 2 was left untreated while group 3 to 6 were treated post-envenoming with 0.2 mL of polyvalent antivenom, 200, 400, and 600 mg/kg of MOE respectively. MOE significantly (*p* < 0.05) normalized the altered haematological indices and blood electrolytes profiles. MOE attenuated venom-induced cellular dysfunctions, characterized by a significant increase in NRF2, and concomitant downregulation of increased antioxidant enzymes (SOD and CAT) activities in the serum and heart of the treated rats. MOE normalized the elevated TNF-α and IL-1β in serum and heart tissues. Furthermore, the IgG titre value was significantly (*p* < 0.5) higher in the envenomed untreated group compared to the MOE-treated groups. Hemorrhagic, hemolytic and coagulant activities of the venom were strongly inhibited by the MOE dose, dependently. Lesions noticed on tissues of vital organs of untreated rats were abolished by MOE. Our findings substantiate the effectiveness of MOE as a potential remedy against EOV toxicities.

**Keywords:** *Moringa oleifera*; antivenom; *Echis ocellatus*; inflammation

## 1. Introduction

Snakebite-related pathology affects as many as 2.7 million people annually, most who live in remote, poorly developed and marginalized tropical communities of the world [1]. Snakebites accounts for annual death ranging from 81,000 to 138,000, while over 400,000 surviving victims suffer permanent psychological and physical disabilities [1,2]. Consequently, envenoming from snakebites is thus a priority among neglected tropical disease by the World Health Organization [3,4]. In Nigeria, the incidence of snake envenoming in remote communities and some parts of urban areas has been on the rise [5]. The upsurge in cases of snake bites might be due to the unique ecosystem, most especially in the North-Eastern and central parts of Nigeria, where the vegetation favours breeding and multiplication of snakes [5].

*Echis ocellatus* (Carpet viper) is regarded as the most medically significant species of viper in West Africa. Its envenoming is poisonous and life-threatening, and accounts for more death than all other African species combined [6,7]. The venom of *E. ocellatus* is highly toxic, consists primarily of peptides and proteins majorly; snake venom metalloproteinases (SVMPs) possess hemorrhagic, nephrotoxic, cardiotoxic and anti-coagulant effects [8].

Thus, envenoming by *E. ocellatus* is responsible for several clinical complications of severe systemic and local pathologies, such as swelling, blistering, necrosis and haemorrhages [9].

Furthermore, the venom of vipers is known to elicit specific immunologic responses in vitro and in vivo or activate inflammatory responses mediated by some cytokines [10,11]. Studies have also shown that toxins in snake venoms generate reactive oxygen species (ROS) associated with inflammatory responses. These free radicals ROS are directly involved in oxidative stress and play a significant role in venom-induced toxicities [12]. Induction of oxidative stress could, in turn, activate a transcription factor known as nuclear factor erythroid 2-related factor 2 (NRF2)—a protein that is known to modulate the expression of antioxidant proteins to the cell protection against oxidative damage triggered by injury and inflammation [13]. Several in vivo and in vitro studies have reported various pathophysiological effects of viper envenoming with concerns focused on *E. ocellatus* [9,14,15].

Over the years, researchers have made several efforts to develop effective therapies for the treatment of snakebites, but to date, antivenom remains the only definitive treatment for snake envenoming. Antivenom has limitations associated with acute and delayed reactions such as anaphylactic and serum sickness reactions combined with its ineffectiveness against local effects after snake envenoming [16,17]. These effects develop very rapidly after snake envenomation, making neutralization by antivenom difficult, especially if treatment is delayed due to either late access to hospital or scarcity of antivenom [18]. All these and more, place a constraint on the use of antivenom in treating snake envenoming. Development of an effective therapy that can overcome drawbacks of current snakebite treatment is, therefore, a clinical priority. Natural means of combating snakebites relies on the use of plants with anti-ophidic effects, with little or no worry about limitations currently associated with antivenom [19].

In the phytotherapy of snakebites, effectiveness of *Moringa oleifera* (Lam) has been ascertained in neutralizing snake venoms of different species [19,20]. However, limited information exists on efficacy of *M. oleifera* on the treatment of complications arising from envenoming by *E. ocellatus*. Accordingly, this study evaluated the bioactivities of ethanol crude extract of *M. oleifera* leave against *E. ocellatus* venom-induced toxicities in vivo and in vitro.

## 2. Materials and Methods

### 2.1. Snake Venom and Antivenom

*E. ocellatus* venom was procured from the Department of Veterinary Pharmacology and Toxicology, Ahmadu Bello University Zaria, Nigeria. A polyvalent antivenom (EchiTAb-Plus ICP) produced at Instituto Clodomiro Picado, University of Costa Rica, Costa Rica was used as standard control treatment for this study.

### 2.2. Ethical Consideration

Animal experiments was approved by the University of Ibadan-Animal Care and Research Ethics Committee (UI-ACUREC), with Assigned number: UI-ACUREC: 18/0108 per the guidelines of the National Code for Health Research Ethics. All animal experiments complied with the National Research Council's publication on guide for the care and use of laboratory animals [21].

#### 2.2.1. Experimental Animals

Eighty male albino Wistar rats weighing 120–160 g used for this study were procured from the animal house of the Department of Zoology, University of Ibadan, Nigeria. Animals were housed in well-ventilated, pathogen-free cages at room temperature (27 °C) and allowed to acclimatize for 14 days under standard conditions (12 h light-dark cycles) before the study commenced. Rats were fed with standard pelleted feed and distilled water were provided *ad libitum*.

2.2.2. Evaluation of Lethality Dose (LD$_{50}$) of *E. ocellatus* Venom

Lethal concentration dose (LD$_{50}$) of the venom was done using a published method [22] with slight modifications. Exactly, 20 rats were randomly divided into four groups (*n* = 5) and injected intraperitoneally (i.p) with varying doses of *E. ocellatus* venom. Group 1 served as control and was injected with 0.2 mL of saline while Group 2 to 4 received 0.1 mL of 0.14, 0.28 and 0.56 mg/kg$^{-1}$ of the venom dissolved in 1 mL of normal saline respectively. The LD$_{50}$ of the venom was calculated with confidence limit at 50% probability using probity analysis of death occurring within 48 h after the venom administration.

*2.3. Collection, Identification and Extraction Procedure of Plant Materials*

*M. oleifera* leaves were freshly harvested from Botanical Garden of the University of Ibadan, and identified in the Herbarium of Department of Botany, University of Ibadan (voucher number UIH-224601). Leaves were air-dried at room temperature for two weeks and blended into fine powder using a super-master electric blender (Model no: SMB-2977). Approximately 3 kg of powdered leaf was extracted in 4 L of absolute ethanol for 72 h using the cold maceration method [23]. The content was then filtered and concentrated using a rotary evaporator at 40 °C.

*2.4. Anti-Ophidic Experiment*
Experimental Design, Envenoming and Treatment Procedures

Thirty male albino rats were randomly selected into six groups (*n* = 5). Group 1 was injected with saline (normal control). Group 2 to 6 were injected intraperitoneally with 0.2 mL of 0.22 mg/kg$^{-1}$ (LD$_{50}$) of *E. ocellatus* in 1 mL of normal saline. Group 2 was not treated post-envenomation (venom control). Group 3 was treated with 0.2 mL of polyvalent antivenom (EchiTab ICP-Plus) intravenously while Groups 4, 5 and 6 were administered orally with (0.2 mL) of 200, 400, and 600 mg/kg$^{-1}$ of *M. oleifera* ethanol extract dissolved in normal saline, respectively. Treatment of envenomed rats commenced 1 h post-envenomation for 7 consecutive days and mortalities were recorded. After seven days of treatment, blood samples were collected from rats via the abdominal veins into anticoagulant bottles for biochemical and haematological analyses, and plain bottles for immunomodulatory assay. The experimental rats were then sacrificed according to guides [24]. Brain, heart and kidneys of rats were harvested for biochemical and histopathological studies.

*2.5. Measurement of Haematological Indices*

The whole blood samples were analyzed for packed cell volume (PCV), white blood cell (WBC) red blood cell (RBC), white blood counts, differentials, haemoglobin level and platelet counts [25].

*2.6. Blood Chemistry Analysis*

Potassium and sodium concentrations were determined using Teco Diagnostics Reagent Kits [26,27] while blood chloride was measured using Agappe diagnostic kits [28].

*2.7. Antioxidant Enzyme Profile*

Heart tissues and serum were analyzed for superoxide dismutase activity (SOD) [29], catalase activity [30] and levels of malondialdehyde [31].

*2.8. Measurement of Inflammatory Cytokines in Serum and Heart Tissues*
2.8.1. Tissue Preparation

Frozen heart tissue samples were homogenized in a 1.5 mL RIPA buffer (25 mM Tris-HCl, 150 mM NaCl, 1% NP-40, 1% sodium deoxycholate, 0.1% SDS pH = 7.6) supplemented with Protease inhibitors at 4 °C. The homogenate was incubated on ice for 30 min and then centrifuged at 10,000× *g* for 30 min at 4 °C. Following centrifugation, the supernatants were transferred to labelled Eppendorf and stored at −80 °C for cytokine measurement.

### 2.8.2. Measurement of Pro-Inflammatory Cytokines and NRF2 Levels in Serum and Heart Tissue

Quantitative measurement of the level of cytokines was performed using Mini Enzyme-Linked Immunosorbent Assay (ELISA) Development Kits (Peprotech). A total of 96-well plates were set up according to the manufacturer's instructions and read using an ELISA plate reader at 405 nm with 650 nm as the correction wavelength. Concentrations (pg/mL) of the serum and heart NRF2, TNF-$\alpha$, and IL-1ß were estimated [32].

### 2.9. Anti-Immunoglobulin G Assay

Briefly, 1 mg/mL of *E. ocellatus* venom diluted in coating buffer (PBS) was used to coat forty-eight well plates (Maxisorp, NUNC, Søborg, Denmark) and was left overnight at 4 °C. The plates were washed 3times with 200 μL of PBS containing 0.05% Tween 20 (PBS-Tween) and blocked for 1 h with 5% fat-free dried milk (Marvel, Nottingham, UK) at 37 °C. Individual sera from immunized animals were diluted 1:500 with 5% milk and applied, in duplicates, to the plates for 2 h at 37 °C. The wells were then washed as above, and 100 mL of a 1:500 dilution of Goat anti-rat IgG (H + L) horseradish peroxidase conjugate (Merck KGaA, Darmstadt, IL, USA) was added. Plates were then incubated for 2 h at 37 °C before rewashing. Chromogenic substrate, 2,20-azino-bis (2-ethylbenzthiazoline-6-sulphonic acid) tablets (Sigma, Poole, UK), were then added and the plates developed at 37 °C for 30 min in phosphate citrate buffer (pH 4.0) containing 0.015% hydrogen peroxide. The OD was read at 405 nm using a model 450 microplate reader (BioRad, Hemel Hempstead, UK).

### 2.10. In Vitro Assays

### 2.10.1. Anti-Hemorrhagic Test

The method of Theakston and Reid [33] was adopted with slight modifications. Thirty albino rats were divided randomly into six groups (*n* = 5). Group 1 was injected intradermally, with 0.2 mL of saline (normal control), while Group 2 was injected with 0.2 mL of $LD_{50}$ of the venom dissolved in 1 mL of saline (venom control). Exactly 0.2 mL of 0.22 mg/kg ($LD_{50}$) of the venom dissolved in 1 mL of normal saline was added to 0.2 mL of different concentrations (200, 400, 600 mg/kg$^{-1}$ in 1 mL of normal saline) of *M. oleifera* extract and 0.2 mL of antivenom in separate in tubes. The mixtures were incubated at 37 °C for 1 h then, 0.2 mL of the mixture was injected intradermally into rats in Group 3 to 6 respectively. The hemorrhagic lesion was measured and recorded after 3 h.

### 2.10.2. Anti-Hemolytic Assay

This assay was determined based on the method of Gomes and Pallabi [34] using 20 mL of citrated bovine blood. Erythrocytes were washed with 5mL of saline (0.9%) by centrifugation (2400 rpm) for 10 min 10 times to obtain free packed cells. After repeated washings with saline, a 10% cell suspension was prepared. 0.2 mL of $LD_{50}$ of the venom (0.22 mg/kg/mL dissolved in 1 mL of saline) was mixed with 2 mL of 10% cell suspension and 0.2 mL of various dilutions of the extract (200, 400, 600 mg/mL) and polyvalent antivenom (0.2 mL) in separate tubes respectively. The venom control tube contained 0.2 mL of venom and 10% cell suspension. Mixtures were incubated at 37 °C for 60 min. The reaction was terminated by adding 3 mL of chilled phosphate buffer saline (PBS pH 7.2). The tubes were centrifuged at 2400 rpm for 10 min and the absorbance of the supernatant was measured at 540 nm. Supernatant of tube treated with 3 mL distilled water was taken as 100% lyses and serves as control.

### 2.10.3. Anticoagulant Assay

This activity was assayed using citrated bovine plasma [34]. Six tubes were arranged in a test tube rack, test-tube 1 contained 0.2 mL of normal saline (Normal control). Exactly, 0.2 mL of $LD_{50}$ of the venom (0.22 mg/kg/mL dissolved in 1 mL of normal saline) was put in test tubes 2 to 6. Test tube 2 served as the Venom control. 0.2 mL of bovine citrated plasma was added to all the test tubes. However, 0.2 mL of various dilutions of

*M. oleifera* extract (200, 400, 600 mg/mL dissolved in 1 mL of normal saline) and 0.2 mL of polyvalent antivenom was added to test tubes 3 to 6, respectively. Then 0.2 mL of $CaCl_2$ was added to all the test tubes, and the mixtures were incubated at 37 °C and clotting times were recorded.

### 2.11. Histopathological Studies

This was carried out using conventional techniques of paraffin-wax sectioning and hematoxylin-eosin staining [35].

## 3. Statistical Analysis

Data were analyzed using Statistical Package for Social Sciences (SPSS version 16) software produced by IBM Corp. Ltd., (Portsmouth, UK).Values were expressed as mean ± standard deviation (S.D). Comparison between different groups was done using one-way Analysis of Variance (ANOVA) followed by the Duncan Multiple Range Test (DMRT) with $p < 0.05$ considered significant.

## 4. Results

### 4.1. Lethal Dose Test of E. ocellatusVenom

The estimated lethal dose ($LD_{50}$) of the venom in this study was 0.22 mg/kg$^{-1}$. Mortality was not recorded in the normal control group. However, in group 2 and 3 envenomed with 0.14 and 0.28 mg/kg$^{-1}$ of *E. ocellatus* venom, 40% mortalities were recorded, while 0.56 mg/kg$^{-1}$ of the venom caused 80% mortality of the animals.

### 4.2. Mortality and Behavioural Responses of Experimental Rats

Death was not recorded in the normal control group and envenomed group treated with antivenom and 600 mg/kg of *M. oleifera* extract. However, 40% mortality was observed in venom control and envenomed group treated with 200 mg/kg of the *M. oleifera* extract, while the group treated with 400 mg/kg of *M. oleifera* extract recorded 20% mortality. Furthermore, one-hour post envenoming rats showed signs of dizziness with reduced activity around their cages. Envenomed rats fed less with increased water consumption.

### 4.3. Effects of Treatment with M. oleifera Extract on Haematological Parameters of Envenomed Rats

The packed cell volume (PCV), red blood cell (RBC) and hemoglobin concentration (Hb) of the venom control were significantly ($p < 0.05$) lower when compared with normal control rats. Dose-dependent increments ensued significantly ($p < 0.05$) in PCV, RBC, Hb, and MCV levels of envenomed rats treated with 200, 400 and 600 mg/kg *M. oleifera* extract respectively. Platelet counts increased markedly in venom control, but were reduced dose-dependently in envenomed and treated groups (Table 1).

**Table 1.** Effect of *M. oleifera* extract on the haematological parameters of the envenomed treated rats.

| Parameters | GROUP 1 | GROUP 2 | GROUP 3 | GROUP 4 | GROUP 5 | GROUP 6 |
|---|---|---|---|---|---|---|
| PCV (%) | 46.00 ± 2.00 [e] | 40.33 ± 0.58 [d] | 53.00 ± 0.00 [b] | 48.00 ± 1.00 [a] | 58.00 ± 1.00 [c] | 63.00 ± 2.00 [b] |
| Hb (g/dl) | 15.23 ± 0.82 [b] | 13.43 ± 0.06 [c] | 15.37 ± 0.15 [b] | 17.70 ± 0.20 [a] | 17.80 ± 0.10 [a] | 19.80 ± 0.10 [b] |
| RBC (cell/L) | 7.92 ± 0.35 [b] | 6.77 ± 0.08 [c] | 8.62 ± 0.08 [b] | 8.15 ± 0.02 [b] | 9.83 ± 0.01 [a] | 9.75 ± 0.05 [a] |
| PLATELET ($\times 10^5$ cell/L) | 1.31 ± 0.20 [c] | 1.44 ± 0.40 [b] | 1.72 ± 0.30 [a] | 1.57 ± 0.10 [b] | 1.29 ± 0.01 [c] | 1.44 ± 0.29 [b] |
| MCV (*fl*) | 58.1 ± 1.21 [c] | 59.5 ± 0.22 [a] | 61.5 ± 0.52 [a] | 58.9 ± 1.43 [ab] | 59.0 ± 1.12 [b] | 64.6 ± 1.81 [a] |
| MCH (*pg*/cell) | 19.2 ± 0.21 [a] | 19.8 ± 0.11 [a] | 20.2 ± 0.21 [b] | 19.3 ± 0.22 [a] | 18.1 ± 0.13 [b] | 20.3 ± 0.22 [a] |
| MCHC (g/dl) | 33.1 ± 0.70 [a] | 33.3 ± 0.62 [a] | 32.8 ± 0.32 [ab] | 32.7 ± 1.11 [ab] | 30.7 ± 0.40 [c] | 31.5 ± 1.11 [b] |

Values are expressed as mean ± S.D ($n$ = 3). Mean with similar superscript on the same column are not significantly different ($p < 0.05$). Hb: hemoglobin, RBC: red blood cell, MCV: mean corpuscular volume, MCH: mean corpuscular haemoglobin, MCHC: mean corpuscular hemoglobin concentration. Group 1: normal control, Group 2: venom control, Group 3: envenomed rats treated with 0.2 mL of polyvalent antivenom, Group 4: envenomed rats treated with 200 mg/kg$^{-1}$ of the extract, Group 5: envenomed rats treated with 400 mg/kg$^{-1}$ of the extract, Group 6: envenomed rats treated with 600 mg/kg$^{-1}$ of the extract.

White Blood Cell and Differentials of the Envenomed Treated Rats

The white blood cell (WBC), lymphocytes (LYM), neutrophil (NEUT), monocytes (MONO), and eosinophil (EO) levels of the venom control group was significantly ($p < 0.05$) lower compared to normal control rats, and all envenomed treated groups. Nevertheless, these parameters increased markedly in treated groups; slightly by the antivenom, but most significantly by the extract. There was no significant difference among the different doses of extract (Table 2).

**Table 2.** White blood cell and differentials of envenomed rats after treatment with ethanol leaf extract of *M. oleifera*.

| Parameters | GROUP 1 | GROUP 2 | GROUP 3 | GROUP 4 | GROUP 5 | GROUP 6 |
|---|---|---|---|---|---|---|
| WBC ($\times 10^3$ cell/L) | 6.5 ± 0.46 [a] | 5.30 ± 0.10 [c] | 5.97 ± 0.33 [ab] | 6.7 ± 0.05 [a] | 6.0 ± 0.05 [ab] | 6.6 ± 1.00 [a] |
| LYM (%) | 76.0 ± 3.01 [b] | 65.0 ± 1.00 [c] | 75.0 ± 1.00 [b] | 64.3 ± 0.60 [c] | 75.0 ± 1.00 [b] | 83.0 ± 1.00 [a] |
| NEUT (%) | 32.3 ± 3.50 [a] | 23.0 ± 1.00 [c] | 22.0 ± 2.00 [c] | 30.0 ± 1.00 [a] | 28.0 ± 1.00 [ab] | 32.7 ± 0.61 [a] |
| MONO (%) | 1.7 ± 0.62 [ab] | 1.0 ± 0.00 [c] | 1.7 ± 0.60 [ab] | 3.0 ± 1.00 [a] | 3.0 ± 0.00 [a] | 2.3 ± 0.62 [ab] |
| EO (%) | 2.0 ± 1.00 [b] | 1.0 ± 0.00 [c] | 2.0 ± 1.00 [b] | 2.0 ± 0.00 [b] | 2.0 ± 0.00 [b] | 2.7 ± 0.60 [a] |

Values are expressed as mean ± S.D, ($n = 3$). Mean with similar superscript on the same column are not significantly difference ($p < 0.05$). WBC: white blood cell, LYM: lymphocytes, NEUT: neutrophils, MONO: monocytes, EO: eosinophils. Group 1: normal control, Group 2: venom control, Group 3: envenomed rats treated with 0.2 mL of polyvalent antivenom, Group 4: envenomed rats treated with 200 mg/kg$^{-1}$ of the extract, Group 5: envenomed rats treated with 400 mg/kg$^{-1}$ of the extract, Group 6: envenomed rats treated with 600 mg/kg$^{-1}$ of the extract.

### 4.4. Effect of Treatment with M. oleifera Extract on Blood Electrolytes of Envenomed Treated Rats

The levels of Cl$^-$ and Ca$^{2+}$ significantly ($p < 0.05$) decreased, whereas no significant difference ($p < 0.05$) exists in Na$^+$ and K$^+$ levels of the venom control group when compared with the normal control group. Treatment with *M. oleifera* extract caused an increase in Cl$^-$ (Figure 1b) and Ca$^{2+}$ (Figure 1d) levels, while there were increases in the Na$^+$ (Figure 1a) and K$^+$ (Figure 1c) as well. The group treated with polyvalent antivenom did not differ from the venom control group (Figure 1).

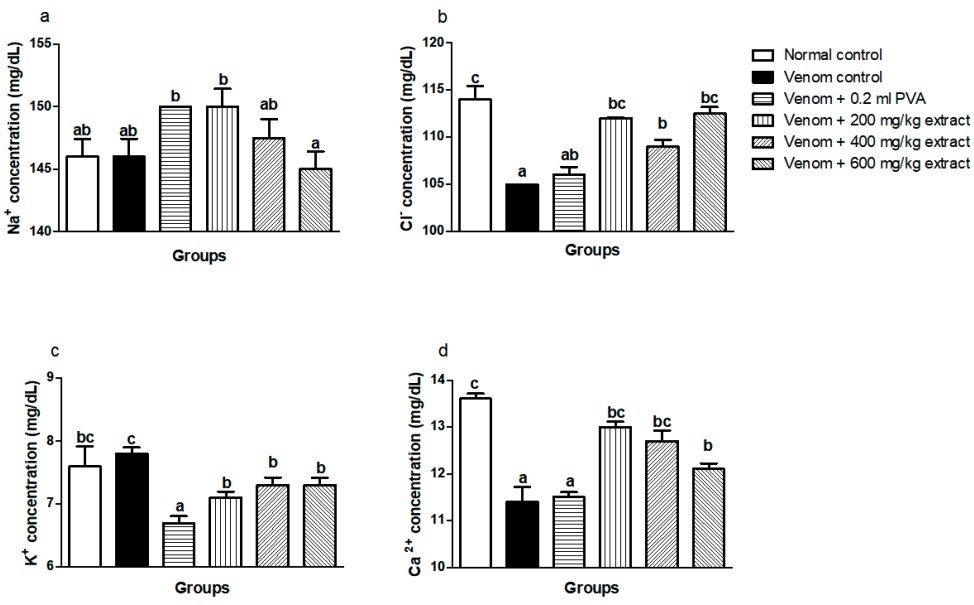

**Figure 1.** Effects of *M. oleifera* ethanol extract on electrolytes levels of rats challenged with *E. ocellatus* venom. Data are expressed as mean ± SD ($n = 3$). Bars with different letters are statistically distinct ($p < 0.05$). (**a**) Na$^+$ concentration; (**b**) Cl$^+$ concentration; (**c**) K$^+$ concentration; (**d**) Ca$^{2+}$ concentration. PVA: polyvalent antivenom; Na$^+$: sodium ion; K$^+$: potassium ion; Cl$^-$: chloride ion; Ca$^{2+}$: calcium ion.

### 4.5. Effects of M. oleifera Extract on Serum and Heart Oxidative Stress Index, Antioxidant Enzymes Activity and Nrf2 Expression of Envenomed Treated Rats

Envenomed untreated groups showed significant ($p < 0.05$) increase in serum MDA level, SOD and CAT enzyme activity when compared to normal control and envenomed rats treated with extract and antivenom, whereas dose-graded reductions in serum MDA level (Figure 2a) and CAT (Figure 2c) enzyme activity occurred in *M. oleifera*-treated rats. There was a significant ($p < 0.05$) reduction in serum SOD enzyme activity of envenomed treated rats which was not dose dependent (Figure 2b). Furthermore, treatment with *M. oleifera* caused a remarkable upregulation of serum NRF2 expression in envenomed treated rats regardless of the dose when compared with venom control (Figure 2d).

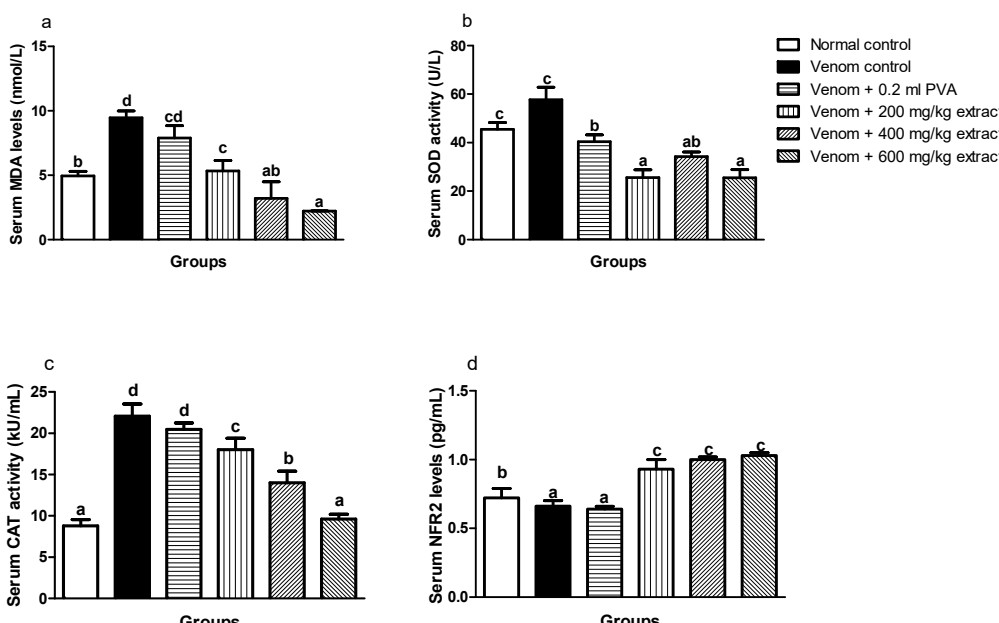

**Figure 2.** Effects of *M. oleifera* extract on MDA (**a**), SOD (**b**), and CAT (**c**) activities, and Nrf2 levels (**d**) in the sera of rats challenged with *E. ocellatus* venom. Data are expressed as mean $\pm$ SD ($n = 3$). Bars with different letters are statistically distinct ($p < 0.05$). PVA: polyvalent antivenom; CAT: catalase; SOD: superoxide dismutase; MDA: malondialdehyde; NRF2: nuclear factor erythroid 2–related factor 2.

Levels of MDA in the heart significantly ($p < 0.05$) increased in envenomed untreated rats when compared to normal control rats (Figure 3a). Similarly, SOD and CAT enzyme activities in the heart increased significantly ($p < 0.05$) in the venom control group compared to normal control rats (Figure 3b,c). However, *M. oleifera* extract showed significant improvements by decreasing MDA levels in the heart which was in a dose dependent manner (Figure 3a). Also, the extract significantly ($p < 0.05$) lowered CAT and SOD enzyme activities at higher doses (Figure 3b,c). Nrf2 expression in the heart significantly ($p < 0.05$) decreased in venom control compared to normal control (Figure 3d). Interestingly, *M. oleifera* extract showed better improvements with significant increase in expression of Nrf2 and this effect was dose dependent. Polyvalent antivenom appears not to produce any significant effect on Nrf2 expression.

### 4.6. Effects of M. oleiferaExtract on Serum and Heart Tumor Necrosis Factor-Alpha (TNF-α) and Interleukin 1–Beta (IL1-β) Expression in Envenomed Treated Rats

A significant increase ($p < 0.05$) was observed in TNF-α levels in serum and heart of venom control rats compared to normal control. However, *M. oleifera* extract significantly ($p < 0.05$) decreased levels of TNF-α in envenomed treated rats (Figure 4a,b). Also, the same trend occurred in the serum and heart of envenomed untreated rats as *M. oleifera* effectively decreased IL-1β levels in treated rats (Figure 4c,d).

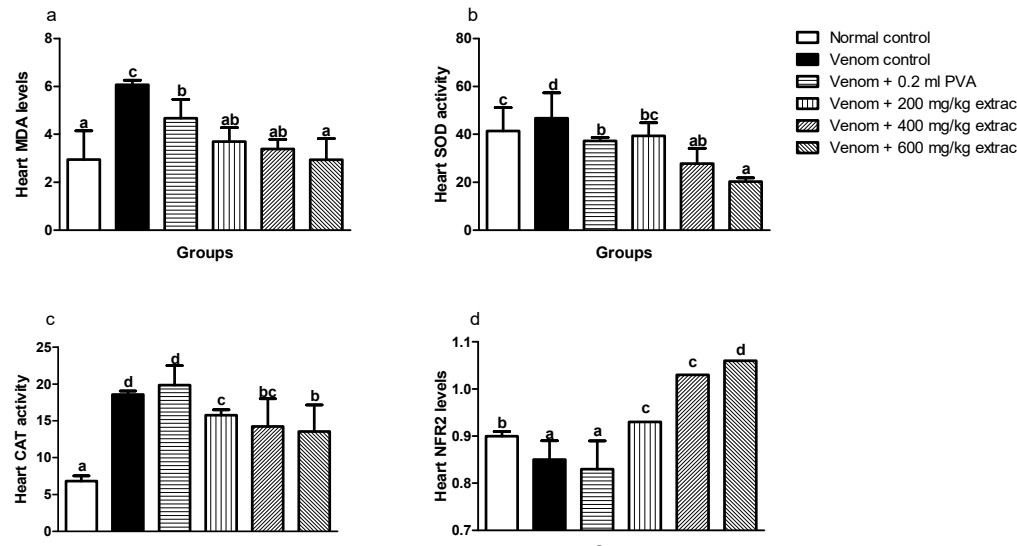

**Figure 3.** Effects of *M. oleifera* extract on MDA (**a**), SOD (**b**), and CAT (**c**) activities, and Nrf2 levels (**d**) in the heart of rats challenged with *E. ocellatus* venom. Data are expressed as mean $\pm$ SD (*n* = 3). Bars with different letters are statistically distinct (*p* < 0.05). PVA: polyvalent antivenom; CAT: catalase; SOD: superoxide dismutase; MDA: malondialdehyde; NRF2: nuclear factor erythroid 2–related factor 2.

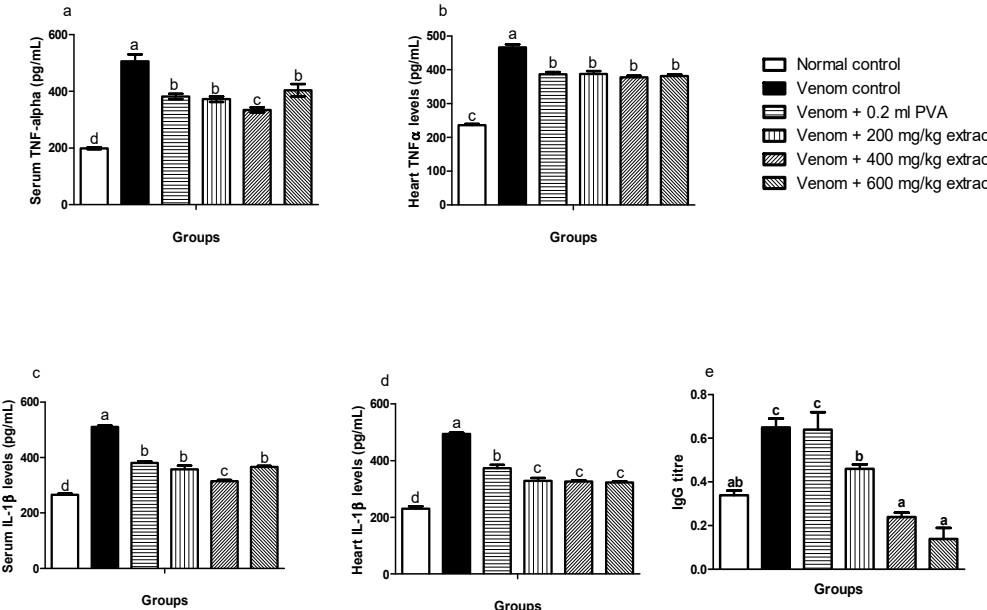

**Figure 4.** Effects of *M. oleifera* extract on TNF: $\alpha$, IL: 1$\beta$ in serum and heart tissues, and immunoglobulin G levels in challenged with *E. ocellatus* venom. Data are expressed as mean $\pm$ SD (*n* = 3). Bars with different letters are statistically distinct (*p* < 0.05). (**a**) serum TNF: $\alpha$; (**b**) heart TNF: $\alpha$; (**c**) serum IL: 1$\beta$; (**d**) heart IL: 1$\beta$; (**e**) serum IgG. PVA: polyvalent antivenom; TNF: $\alpha$-Tumour necrosis factor-alpha; IL: 1$\beta$-interleukin-1 beta; IgG: immunoglobulin G.

### 4.7. IgG Antibody Titre of Envenomed Rats Treated with M. oleiferaExtract

The IgG antibody titre of the envenomed untreated group was significantly (*p* $\leq$ 0.05) higher compared to the normal control group and envenomed treated groups. However, *M. oleifera* extract significantly (*p* $\leq$ 0.05) decreased IgG antibody titre in envenomed treated groups, which were dose dependent. Antivenom did not show any significant effects when compared to venom control (Figure 4e).

*4.8. In Vitro Studies*

4.8.1. Anti-Hemorrhagic Activity of *M. oleifera* Extract

Hemorrhagic foci were not observed in the normal control group, whereas venom control showed 100% haemorrhage. However, antivenom showed 58% inhibition of hemorrhage induced by *E. ocellatus* venom, while *M. oleifera* extract dose dependently inhibited hemorrhagic activities of the venom (Table 3).

**Table 3.** Anti-hemorrhagic activity of *M. oleifera* extract.

| Treatment Groups | Percentage Inhibition (%) |
|---|---|
| Group 1 | - |
| Group 2 | 0 (No inhibition) |
| Group 3 | 58.00 $\pm$ 8.00 [c] |
| Group 4 | 72.00 $\pm$ 7.00 [b] |
| Group 5 | 84.00 $\pm$ 4.00 [ab] |
| Group 6 | 95.00 $\pm$ 11.72 [a] |

The superscript along the column was used to indicate the level of significance. i.e., values with different letters along the column are statistically distinct $p < 0.05$. Data are expressed as means $\pm$ S.D. of three individual experiments ($n = 3$). Legend: Group 1: rats injected with 0.2 mL of normal saline, Group 2: rats injected with 0.2 mL of the venom. Group 3: rats injected with 0.2 mL of venom/200 mg/kg extract, Group 4: rats injected with 0.2 mL venom/400 mg/kg extract, Group 5: rats injected with 0.2 mL of venom/600 mg/kg extract, Group 6: rats injected with 0.2 mL of venom/polyvalent antivenom.

4.8.2. Anticoagulant Studies

Citrated blood samples of normal control clotted at 48 s, whereas venom control samples clotted at 289 s. All extract treated samples clotted between 65 and 210 s. Ahigh dose (600 mg/kg) of *M. oleifera* extract was most effective as plasma clotted at 65 s. Also, antivenom samples clotted at 58 s (Figure 5).

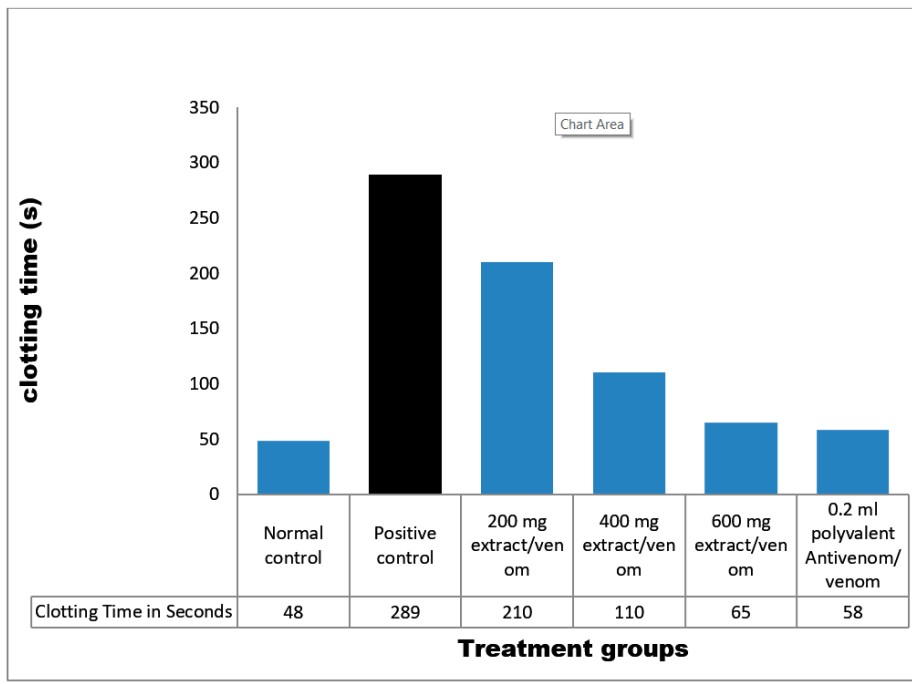

**Figure 5.** Coagulant activity of *M. oleifera* extract against *E. ocellatus* venom. Data are expressed as means $\pm$ S.D. of three individual experiments ($n = 3$). Group 1: distilled water/citrated plasma (normal control), Group 2: venom/citrated plasma (venom control), Group 3: venom/citrated plasma/100 mg/kg extract, Group 4: venom/citrated plasma/200 mg/kg extract, Group 5: venom/citrated plasma/300 mg/kg extract, Group 6: venom/citrated plasma/0.2 mL polyvalent antivenom.

### 4.8.3. Antihemolytic Activity of *M. oleifera* Leave Extract

The venom caused remarkable hemolytic effects on bovine erythrocytes but polyvalent antivenom was effective and exhibited 80% antihemolytic effect. Also, various concentrations of *M. oleifera* extract caused a dose-dependent reduction in blood lyses. Samples treated with high dose (600 mg/kg) of extract recorded the strongest inhibition of hemolysis (Table 4).

**Table 4.** Effect of *M. oleifera* extract on hemolytic activity of *E. ocellatus* venom.

| GROUPS | % Inhibition |
|--------|--------------|
| 1 | 0 (No inhibition) |
| 2 | 28.00 ± 4.00 [c] |
| 3 | 80.00 ± 10.00 [b] |
| 4 | 65.00 ± 5.00 [ab] |
| 5 | 71.00 ± 1.00 [b] |
| 6 | 87.00 ± 7.00 [a] |

Values are expressed as mean ± S.D of three samples. The superscript along the column was used to indicate the level of significance. i.e., values with different letters along the column are statistically distinct $p < 0.05$. Group 1: mixture of distilled water/citrated RBCs (normal control). Group 2: mixture of venom/citrated RBCs (positive control). Group 3: mixture of venom/citrated RBCs/100 mg/kg of crude extract. Group 4: mixture of venom/citrated RBCs/200 mg/kg of crude extract. Group 5: mixtures of venom/citrated RBCs/300 mg/kg of crude extract. Group 6: mixtures of venom/citrated RBCs/standard polyvalent antivenom.

### 4.9. Histological Studies

#### 4.9.1. Histopathological Studies of the Brain

The brain of normal control rats appeared normal without any histological impairment, whereas the brain of the venom control rats revealed a markedly widespread vacuolation, infiltration of the inflammatory cells, combined with foci of degeneration of neurons. However, slides of envenomed rats treated with *M. oleifera* extract showed fairly-normal neurons, vacuolation of the neuropil and a mild thickening of the meninges. Photomicrograph of antivenom-treated rats showed neurons with congestion of interstitial blood vessels and no significant damage (Figure 6).

#### 4.9.2. Histopathological Studies of the Heart

The heart of the normal control group showed no visible lesion while the heart of venom control rats revealed a multiple foci of thinning and attrition of cardiomyocyte fibers, infiltration of inflammatory cells and presence of epicardia hemorrhagic foci. Envenomed rats treated with antivenom showed normal cardiac muscle fibers with a marked congestion of the coronary blood vessels. The heart of envenomed rats treated with different doses of *M. oleifera* extract showed a mild pallor of the cardiomyocytes and there was no remarkable histological impairment (Figure 7).

#### 4.9.3. Histopathological Studies of the Kidney

The kidneys of control rats showed normal tubules, glomerular capillaries and renal interstitial blood vessels. However, slides of kidneys of untreated envenomed rats revealed a local extensive foci of marked flattening of tubular epithelial cells, giving the tubules a cystic dilated appearance. Histological slides of envenomed rats treated with antivenom showed a moderate congestion of glomerular capillary tufts and interstitial blood vessels whereas a local extensive foci of marked flattening epithelium of tubules and extensive marked congestion of renal interstitial blood vessels were noticed in the kidneys of groups treated with *M. oleifera* extract (Figure 8).

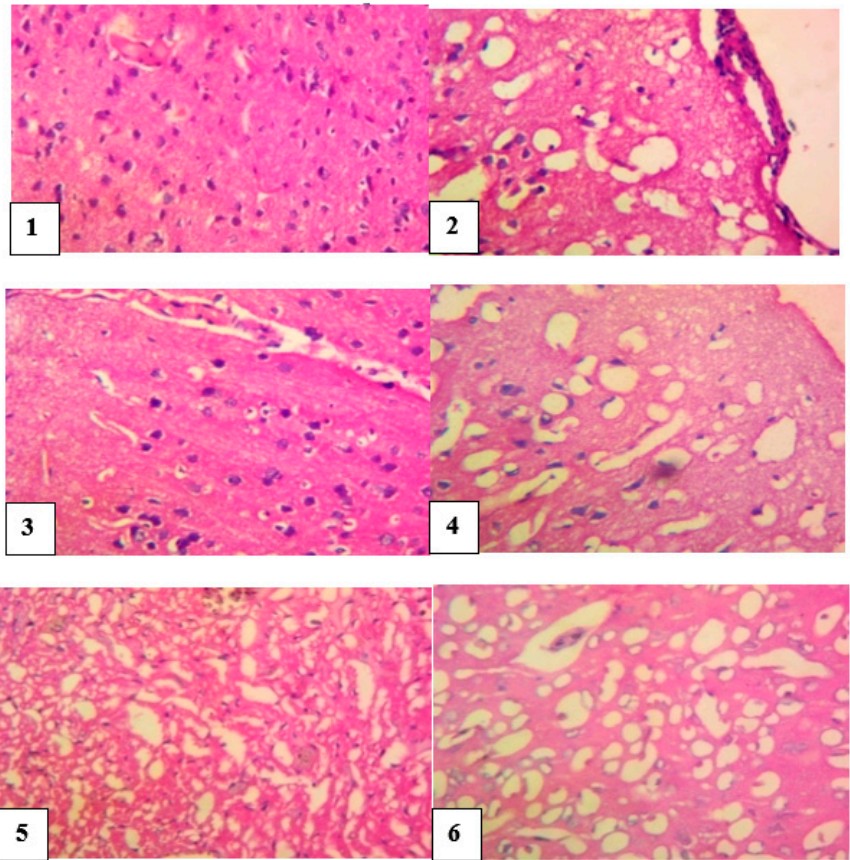

**Figure 6.** Histological evaluation of the brain of envenomed treated rats (×400).Group **1** (normal control): neurons appeared normal, Group **2** (venom control): vacuolation and inflammatory cells infiltration, foci of degeneration, and necrosis of neurons, Group **3** (antivenom control): neurons appeared fairly normal, Group **4** (venom/200 mg/kg extract): normal appearance of neurons, Group **5** (venom/400 mg/kg extract): neurons appear fairly normal, Group **6** (venom/600 mg/kg extract): neurons appeared normal.

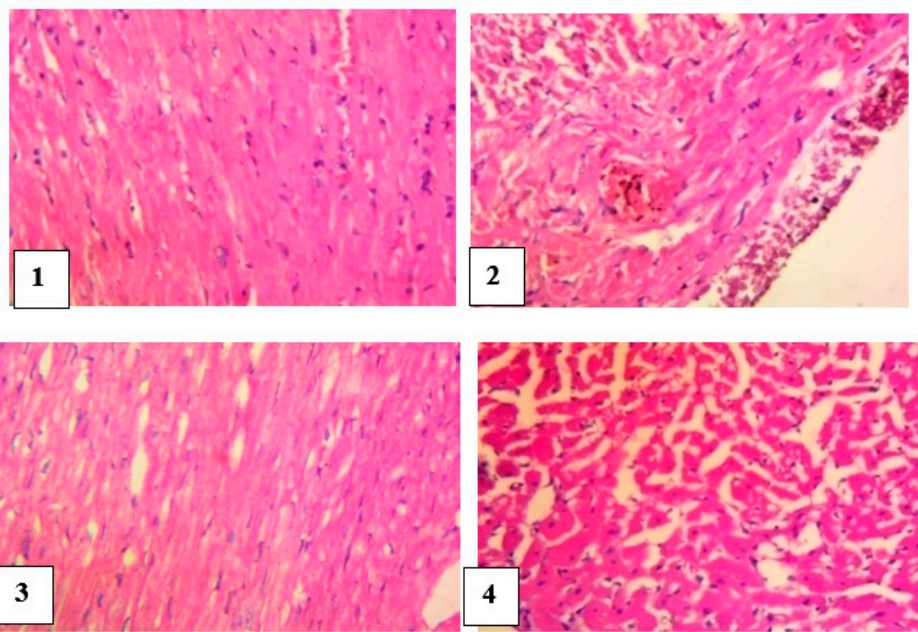

**Figure 7.** *Cont.*

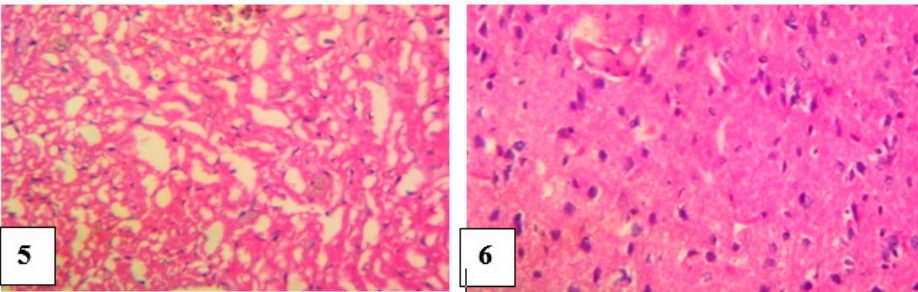

**Figure 7.** Histological examination of the heart of envenomed treated rats (×400) Group **1** (normal control): cardiomyocytes appeared normal, Group **2** (venom control): multiple foci of thinning and attrition of cardiomyocyte and epicardia hemorrhages, Group **3** (antivenom control): extensive foci of mild pallor of the cardiomyocytes, Group **4** (venom/200 mg/kg extract): focus of the cardiomyocytes, Group **5** (venom/400 mg/kg extract): few foci and mild pallor of the cardiomyocytes, Group **6** (venom/600 mg/kg extract):cardiomyocytes appeared normal.

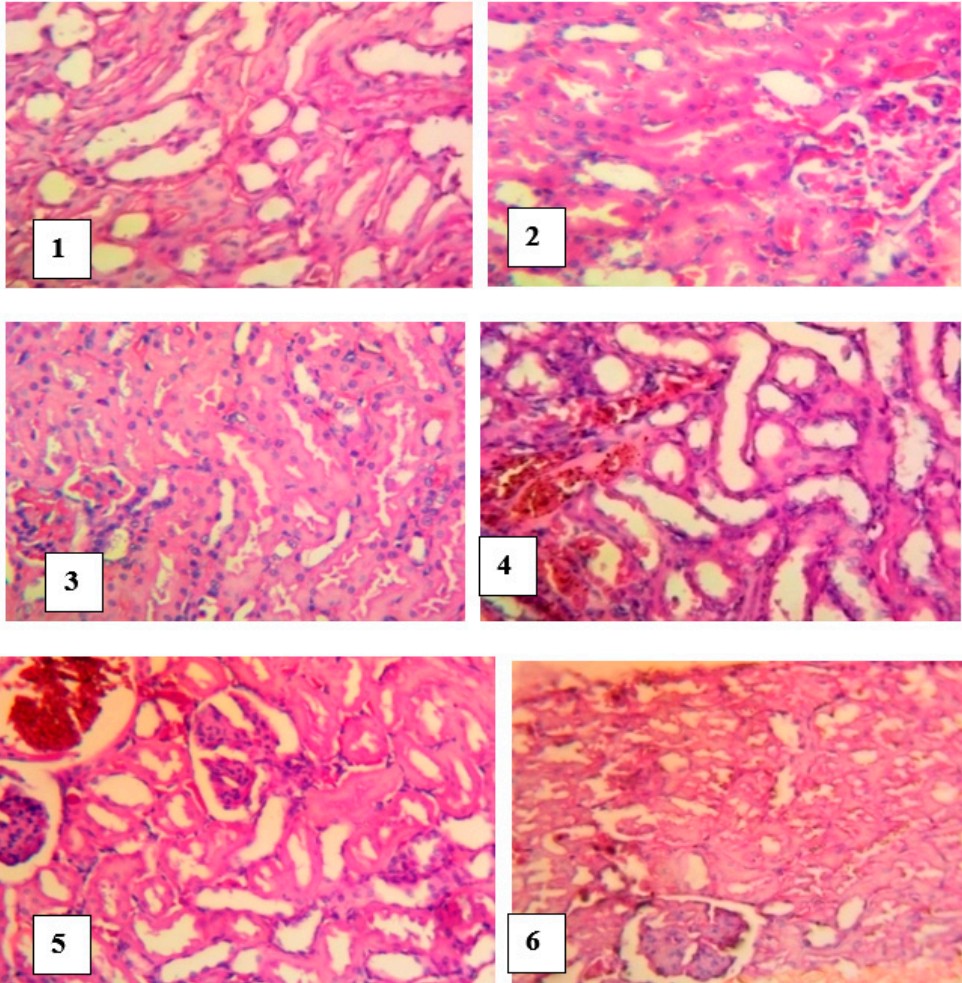

**Figure 8.** Histological examination of the kidneys of envenomed treated rats (×400). Group **1** (normal control): Normal and no foci of flattening of tubular epithelial cells, Group **2** (venom control): a cystic dilated appearance of foci of tubules, Group **3** (antivenom control): tubules appeared normal, Group **4** (venom/200 mg/kg extract): extensive foci of flattening of epithelium of tubules, Group **5** (venom/400 mg/kg extract): focus of marked flattening of epithelium of tubules, Group **6** (venom/600 mg/kg extract): few foci of tubular epithelial cells.

## 5. Discussion

Viper envenoming is considered an occupational health hazard among agricultural workers, farmers and nomads, most especially in local communities in Nigeria [5]. Most rural dwellers depend on medicinal plants for the treatment of snakebite due to limited access to hospital treatment combined with scarcity and high cost of antivenom. A better understanding of venom-induced pharmacology and the development of effective phytotherapy could minimize snakebite mortality and morbidity in the region. This present study thus demonstrated *E. ocellatus* venom-induced toxicities and presents possible amelioration using *M. oleifera* extract, as plants are considered sufficiently available and easy to access if need be.

Viper's venoms are considered poisonous as they contain large number of toxic components. The components showing the major concentrations correspond to zinc-dependent snake venom metalloproteinases (SVMPs), phospholipase-A2 (PLA2s) and serine proteinases [36,37]. SVMP are largely responsible for the degradation of the basement membrane of micro-vessels, with the consequent haemorrhage [38], activation of prothrombin and factor-X [39], thus generating the formation of micro-thrombi and fibrinogen depletion, that is, defibrinogenation [40], and degradation of the extracellular matrix [41], among other effects.

In this study, toxins present in *E. ocellatus* venom expressed various toxic effects in vivo and in vitro. On the other hand, ethanol leaves extract of *M. oleifera* displayed effective antivenom action.MOE has been reported to possess antivenom activity due to the presence of phytochemicals such as the alkaloids, saponins, and flavonoids [19]. High mortality was recorded in untreated rats which, however, reduced upon treatment with *M. oleifera* extract, suggesting that the extract was able to halt the progression of lethal effects of the venom toxins. Results of haematological studies showed that envenoming caused acute anaemia in untreated rats as characterized by reduced PCV and Hb levels. This effect could be attributed to the mechanism of action of viper venoms as they are hemotoxic and cytotoxic. These results corroborated other reports [42]. However, anaemia was reversed after treatment with *M. oleifera* extract with a dose-dependent improvement in the haematological indices, an indication that the extract possesses erythropoietic properties, although anti-anaemic effects of *M. oleifera* has been documented [43]. This observation is in tandem with previous studies [20]. Also, alterations in sodium, chloride, potassium and calcium ion concentrations may be due to homeostasis disturbance following viper bites [44].Furthermore, neutrophil count decreased considerably in the untreated envenomed group. Snake venom has been reported to be cytotoxic to neutrophils [45]. Neutrophils are responsible for clearance of damaged tissues, phagocytosis of the infectious agent and they regulate inflammatory processes. Reduction in neutrophils counts therefore might suggest immunosuppressive activity of the venom. However, treatment with MOE normalized the neutrophil counts comparable to normal control, suggesting its immunomodulatory effect [46,47].

The present study demonstrated that *E. ocellatus* venom can induce oxidative dysfunction as revealed by a reduction in levels of nuclear factor erythroid 2-related factor 2 (Nrf2) in serum and heart tissues of untreated rats. Nuclear factor erythroid-related factor 2 (Nrf2) is a critical antioxidant transcription factor that regulates the expression of many antioxidant and phase II detoxifying enzyme genes, such as haem oxygenase-1 (HO1) and NADP (H): quinine oxidoreductase-1 (NQO1), through the antioxidant response element (ARE). Nrf2 is repressed in the cytoplasm by the Kelch-like ECH associating protein-1 (Keap1) under normal conditions. However, Nrf2 dissociates from Keap1 and translocates to the nucleus to bind to ARE upon oxidative stress [48]. Activation of the Nrf2-ARE pathway has a protective effect against various diseases via antioxidative mechanisms [49]. Reduction in Nrf2 expression in untreated rats is an indication of venom-induced oxidative dysfunction as a result of reactive oxygen species (ROS) leading to oxidative stress. Consequently, this may explain the observed oxidative stress in envenomed untreated rats as activities of the antioxidant enzymes (SOD and CAT) were up-regulated in serum and heart tissues of this

group. However, *M. oleifera* attenuated this effect by up-regulating the Nrf2 expression, as evident in serum and heart tissues of envenomed treated groups.

Findings clearly showed that *E. ocellatus* venom increased the activity of SOD and catalase activity. Superoxide dismutase (SOD) from the front line of defense against reactive oxygen species (ROS)-mediated injury [50]. The induction of enzymatic and non-enzymatic defensive mechanisms on exposure to the venom could be an adaptive or a compensatory response that enables the cells to overcome the damage. High activities of catalase and SOD enzymes in serum and heart tissue of untreated rats confirmed increased free radicals' production and suggested possible damage in oxidant system. Results aligned with other studies on increased activities of SOD and catalase in vital organs of envenomed animals [51]. However, *M. oleifera* leaves extract down-regulated the activities of the antioxidant enzymes which is in consonance with previous studies [20].

To further demonstrate the involvement of oxidative stress in venom-induced toxicity, the levels of Malondialdehyde (MDA) were measured. MDA is a product of lipid peroxidation and has been used as a biomarker of oxidative stress [52–54]. Significant increase in the MDA of serum and heart tissue of untreated rats further confirmed the ability of *E. ocellatus* venom to induce oxidative damage. Also, measuring the production of MDA levels is generally considered a strong indicator of lipid peroxidation [55]. The ability of the extracts to normalize the activity of SOD and CAT as well as reducing the level of MDA may be attributed to its antioxidant properties due to the presence of electron-donating phytochemicals [56] which may have anti-ROS effects.

Cytokines are small, secreted proteins released by cells and have a specific effect on the interactions and communications between cells [57]. Studies have demonstrated the involvement of pro-inflammatory cytokines in the activation of endothelial cells and leukocytes, leading them to express cell surface adhesion molecules using viper venom [58]. Results obtained in this study suggested that *E. ocellatus* venom can stimulate cells to secrete pro-inflammatory cytokines as the venom significantly increases TNF-$\alpha$ and IL-1ß levels in serum and heart tissue of envenomed rats. These results supported earlier studies showing augmented levels of cytokines in sera of experimental animals injected with viper venoms [59]. However, *M. oleifera* extract attenuated the expression of pro-inflammatory cytokines in serum and heart of envenomed treated rats. Although, studies have reported anti-inflammatory properties of *M. oleifera* [60,61].

*E. ocellatus* venom stimulated the experimental rats to produce specific antibodies via immunomodulation as the IgG antibody titre significantly increased compared to normal control. IgG is a primary antibody that protects the body from infectious diseases, pathogens, and other foreign bodies [62]. Increased IgG antibody titre in untreated rats may be a physiological response of the immune system to protect against the venom toxins. Also, this observation is an indication that the venom is highly immunogenic and the immunization protocol used in this study is suitable for the production of high titre snake venom antisera. Immunomodulation as a result of venom action has been previously reported [19]. On the other hand, *M. oleifera* extract altered the immune responses produced by the venom as IgG antibody titre was significantly decreased in envenomed treated rats affirming that *M. oleifera* possesses immunomodulatory action as earlier reported [19].

*E. ocellatus* venom induced a direct haemolysis on the bovine erythrocytes in this study. Viperid venom are known to contain phospholipase-A2 (PLA2s) which cause cell damage [63]. In addition, snake venom PLA2s are responsible for hydrolysis of the membrane phospholipids of blood cells and the subcellular organelles. The lysophosphatidylcholine generated by PLA2s is considered as a potent inducer of inflammation and tends to exacerbate the intensity of ROS production at the site of inflammation. The generated ROS can induce a number of molecular alterations in cellular components of blood [51], and in turn, results in blood lysis due to cytotoxic effects of viper venom. However, *M. oleifera* extract countered the hemolytic effects of the venom in a dose-dependent manner. This result support earlier findings on blood lytic activity of viper venom and anti-hemolytic effects of *M. oleifera* extract [20].

The venom of *E. ocellatus* is hemotoxic, and causes haemostasis disturbances such as delay in coagulation, fibrinolysis and haemorrhagic activity. These effects are largely due to the enzymatic action of metalloprotease, a major enzyme present in the venom [64,65]. In this study, recalcification of citrate bovine plasma was prolonged in venom control, which is an indication of venom-induced coagulopathy. The observed increase in clotting time could be as a result of the inhibition of platelet aggregation or the activation of prothrombin. However, clotting time of bovine plasma incubated with polyvalent antivenom and various doses of the plant extract were shortened. This confirms that the extract could restore coagulation as previously documented [20].

Another positive attribute of the extract worthy of note was its anti-haemorrhagic activity against *E. ocellatus* venom. Haemorrhagic foci caused by the venom was significantly decreased based on extract concentration. Studies have attested to the effectiveness of *M. oleifera* extract against haemorrhage induced by snake venoms [19,20]. These findings reflect that the extract of *M. oleifera* leaves may contain an endogenous inhibitor of snake venom haemorrhagic toxins, but this assertion would need further investigations as the underplaying mechanism yet remains elusive.

*E. ocellatus* caused various pathological damage on tissues of vital organs of untreated rats and these effects could be attributed to ROS produced by the venom. ROS are involved in inflammatory responses, thereby affecting the cellular physiology and playing a significant role in the pathological conditions [47,66]. However, *M. oleifera* ameliorated various histopathological defects caused by the venom as observed in slides of treated rats.

Researchers have documented that *M. oleifera* possesses numerous bioactive constituents such as vitamins, phenolic acids, flavonoids, isothiocyanates, tannins, alkaloids and saponins, which are present in significant amounts [67]. The mechanism of venom inhibition by the compounds present in plants was reported to be as a result of free radical formation inhibition through lipoxygenase and cyclo-oxygenase pathway [68] or possession of protein binding and enzyme inhibiting properties [69]. The mechanism of plant compounds against snake envenomation is somehow similar which could be applicable in this study.

## 6. Conclusions

In this study, toxins in *E. ocellatus* displayed various toxic effects as expected. The venom displayed its ability to invoke excessive production of ROS capable of inducing oxidative stress and activating the inflammatory pathways, thus, culminating in organ dysfunction and possible mortality. However, *M. oleifera* extract extenuated these toxic effects. This ameliorative effect is characterized by normalization of the antioxidant enzymes, attenuation of oxidative stress marker, inhibition of hemolysis, coagulation, and haemorrhage. The mechanism of action may be attributed to the enhancement of the antioxidant response system via NRF2 signaling pathways and attenuation of pro-inflammatory cytokines production.

**Author Contributions:** Conceptualization, A.O.A.; data curation, A.S.J.; Formal analysis, A.O.A.; investigation, A.O.A., O.E.A. and E.-O.P.E.; project administration, S.O.A. and B.S.A.; resources, O.E.A.; supervision, E.-O.P.E.; validation, A.S.J.; writing—original draft, B.S.A.; writing—review andediting, A.S.J. All authors have read and agreed to the published version of the manuscript.

**Funding:** This research received no external funding.

**Institutional Review Board Statement:** The study was conducted according to the guidelines of the Declaration of Helsinki, and approved by the Institutional Review Board of University of Ibadan-Animal Care and Research Ethics Committee (UI-ACUREC)(UI-ACUREC: 18/0108).

**Informed Consent Statement:** Not applicable.

**Conflicts of Interest:** Authors declare no conflict of interests.

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
