# Peer review of "Moringa oleifera Extract Extenuates Echis ocellatus Venom-Induced Toxicities, Histopathological Impairments and Inflammation via Enhancement of Nrf2 Expression in Rats"

_pathophysiology, doi:10.3390/pathophysiology28010009_

Round 1
Reviewer 1 Report
This manuscript is focused on observing anti-venom functions of M. oleifera against E. ocellatus venom toxicities. In the manuscript, it was quite interesting that the leaves of M. oleifera significantly inhibited the hemorrhagic and hemolytic activities of E. ocellatus venom. Generally, this observation is important to understand the effectiveness of potential anti-venom substrate against snake envenomation. Although the manuscript is well written and summarized, there are several points which have to be concerned and modified before full recommendation.
(1) Extensive editing of English language and style is recommended.
(2) Since, various components are present in M. oleifera extract, it is hard to find the factor which is related in mechanism of hemorrhagic or hemolytic activities against E. ocellatus venom and this fact attenuates the authors’ opinion. In this manuscript, expression profiles of M. oleifera leaves based on transcriptome analysis might support the importance of components. Therefore, I recommend authors to briefly put additional descriptions for the expression profiles of components such as hemostasis affecting components in M. oleifera leaves in “Discussion section”.
(3) Unify the scientific names in Italic. (Line 14, 94, 316, 340, 556, 561, 562, 563, 565, 576, 582, 605, 649, 651, 653, 655, 665 and 668)
(4) Change ‘in vivo’ into Italic. (Line 73)
(5) Change 3000 g into 3 kg. (Line 107)
(6) Change the letter style ‘Italic’ into normal style. (Line 132-133)
(7) Do not give a space between number and %. (Line 174, 176, 180, 207, 208, 213 and 342)
Author Response
Response to Reviewer 1 Comments
Comment 1
(1) Extensive editing of English language and style is recommended.
Response 1
The written English Language of the manuscript has been extensively improved as suggested.
Comment 2
(2) Since, various components are present in M. oleifera extract, it is hard to find the factor which is related in mechanism of hemorrhagic or hemolytic activities against E. ocellatus venom and this fact attenuates the authors’ opinion. In this manuscript, expression profiles of M. oleifera leaves based on transcriptome analysis might support the importance of components. Therefore, I recommend authors to briefly put additional descriptions for the expression profiles of components such as hemostasis affecting components in M. oleifera leaves in “Discussion section”.
Response 2
A published work Ajisebiola et al. (2020) alluded the bioactivity of MOE against snake venom to the presence of flavonoids, alkaloids, and saponins. This has been included in the discussion section.
(3) Unify the scientific names in Italic. (Line 14, 94, 316, 340, 556, 561, 562, 563, 565, 576, 582, 605, 649, 651, 653, 655, 665 and 668)
Response 3
The scientific names have been italicised as suggested
(4) Change ‘in vivo’ into Italic. (Line 73)
Response 4
The ‘in vivo’ have been italicised as suggested
(5) Change 3000 g into 3 kg. (Line 107)
Response 5
The 3000 g has been changed to 3kg
(6) Change the letter style ‘Italic’ into normal style. (Line 132-133)
Response 6
This has been done
(7) Do not give a space between number and %. (Line 174, 176, 180, 207, 208, 213 and 342)
Response 7
The spaces have been removed
Reviewer 2 Report
Dear authors,
The manuscript related to the effectiveness of MOE as a potential remedy against EOV toxicities is a good work suggesting that medicinal plants can be used for the treatment of snakebite due to limited access of the victims to hospital treatment along with scarcity and high cost of antivenom.
However, some points need to be clarified.
1- Why the authors chose the i.p. route for the administration of the venom? If the authors would like to mimic the snakebite, it is better to use the gastrocnemius route.
2- In the Introduction section, 3rd paragraph, the authors mentioned the NRF2. However, what is the relation of this protein with snakebites? This information is out of context. Please, rewrite this part.
3- The line 67, please change to the anti-ophidic effect. Because the plant does not affect the bite.
4- Line 88: procured?
5- Line 92: What the meaning of "clean water"? Filtered, autoclaved...
6- Line 134: Why the authors chose to measure the cytokines in serum and heart? Why they do not measure in the peritoneal exsudate, the kidney, and the brain? At what time it was determined? If the authors treated the animals at different time intervals, why they do not measure these cytokines in several time intervals? It is well known that they modulate each other.
Do the authors take a look at the urine? What it look like? Clear, dark, yellow?
7- Line 148: Please include the data presentation. The results were presented as pg/mL.
8- Line 149: What the authors would like to immunomodulate? Please, choose a better title for this methodology.
9- Results section: Please clarify this sentence. 40% mortality with 200 mg/kg and 20% with 400 mg/kg?
However, 40 % mortality was observed in 211 venom control and envenomed group treated with 200 mg/kg of the M. oleifera extract 212 while the group treated with 400 mg/kg of M. oleifera extract recorded 20 % mortality.
10- Discussion section needs some improvement.
Please these published papers can help in some parts.
- Toxicon. 2020 Nov;187:188-197. doi: 10.1016/j.toxicon.2020.09.006.
- Mem Inst Oswaldo Cruz. 2005 Mar;100 Suppl 1:181-4. doi: 10.1590/s0074-02762005000900031
- Toxicon. 2018 Apr;145:48-55. doi: 10.1016/j.toxicon.2018.02.046.
- Curr Med Chem. 2014;21(25):2952-79. doi: 10.2174/09298673113206660295
Author Response
Response to Reviewer 2 Comments
POINT1- Why the authors chose the i.p. route for the administration of the venom? If the authors would like to mimic the snakebite, it is better to use the gastrocnemius route.
RESPONSE 1
The intraperitoneal route was used based on previous work. The I,p route will ensure rapid circulation of the venom into the blood. Adeyi et al. (2020). However, the gastrocnemius route will be considered in another study.
POINT 2- In the Introduction section, 3rd paragraph, the authors mentioned the NRF2. However, what is the relation of this protein with snakebites? This information is out of context. Please, rewrite this part.
RESPONSE 2
NRF2 is an antioxidant- response transcription factor that is usually induced in response to oxidative insults. Snake venom might induce oxidative stress via generation of ROS; this might cause inflammation if not checked, hence, the reason for mentioning NRF2.
POINT 3- The line 67, please change to the anti-ophidic effect. Because the plant does not affect the bite.
RESPONSE 3
Anti-snake has been changed to anti-ophidic
POINT 4-Line 88: procured?
RESPONSE 4
Obtained has been changed to procured
POINT 5- Line 92: What the meaning of "clean water"? Filtered, autoclaved...
RESPONSE 5
The water offered to the animals is distilled water.
POINT 6- Line 134: Why the authors chose to measure the cytokines in serum and heart? Why they do not measure in the peritoneal exsudate, the kidney, and the brain? At what time it was determined? If the authors treated the animals at different time intervals, why they do not measure these cytokines in several time intervals? It is well known that they modulate each other.
Do the authors take a look at the urine? What it look like? Clear, dark, yellow?
RESPONSE 6
The cytokine has been measured in the serum to reveal the circulating amount of the cytokines in the blood, the cardiotoxic effect of the venom has been reported (doi: 10.1016/S0377-1237(02)80054-5), thus, we sought to investigate the cytokine level in the heart. However, further studies might focus on the suggested compartments.
We did not investigate the urine.
POINT 7- Line 148: Please include the data presentation. The results were presented as pg/mL.
RESPONSE 7
The unit of ELISA kit used to quantify (pg/ML) has been included.
POINT 8- Line 149: What the authors would like to immunomodulate? Please, choose a better title for this methodology.
RESPONSE 8
The title has been changed to anti-immunoglobulin G assay
POINT 9- Results section: Please clarify this sentence. 40% mortality with 200 mg/kg and 20% with 400 mg/kg?However, 40 % mortality was observed in 211 venom control and envenomed group treated with 200 mg/kg of the M. oleifera extract 212 while the group treated with 400 mg/kg of M. oleifera extract recorded 20 % mortality.
RESPONSE 9
The sentence indicates that, the envenomed rats treated with 200 mg/kg MOE had 40% mortality, whereas, those treated with 400 mg/kg MOE had 20% mortality.
10- Discussion section needs some improvement.
RESPONSE 10
The discussion has been improved
Round 2
Reviewer 2 Report
Dear authors,
I have some minor considerations about the last update.
1- Please take a careful look at the scientific names and put them in Italic.
2- Line 49: Insert a reference about in vitro studies with venom. Can be a revision like this recently published: Zuliani et al., 2020; Or a research article like Nery et al., 2020.
3- Line 51: Insert a reference about toxins and ROS production such as Paloschi et al., 2018 and Zuliani et al., 2005.
4- Line 69: please correct for anti-ophidic change t for d
5- According to MM and table 2, the venom did not induce a neutrophil influx after 7 days of envenomation. But, in Results and Discussion sections the authors mentioned that " neutrophil count decreased considerably in the untreated envenomed group". Please clarify.
6- line 430: change anti-venom for anti-ophidic.
7- line 445: This is not a better reference for this sentence. Change for: Zuliani et al., 2020.
Author Response
Pathophysiology Responses to Reviewers’ comments
Comment 1- Please take a careful look at the scientific names and put them in Italic.
Response
The entire manuscript has been checked paragraph by paragraph, and the scientific names have been put in italics
Comment 2- Line 49: Insert a reference about in vitro studies with venom. Can be a revision like this recently published: Zuliani et al., 2020; Or a research article like Nery et al., 2020.
Response
Zuliani et al. (2020) has been cited as suggested. See line 49, 50, and 57
Comment 3- Line 51: Insert a reference about toxins and ROS production such as Paloschi et al., 2018 and Zuliani et al., 2005.
Response
The two references have been cited at the end of the sentence. See line 52
Comment 4- Line 69: please correct for anti-ophidic change t for d
Response
The ‘t’ has been exchanged with a ‘d’
Comment 5- According to MM and table 2, the venom did not induce a neutrophil influx after 7 days of envenomation. But, in Results and Discussion sections the authors mentioned that " neutrophil count decreased considerably in the untreated envenomed group". Please clarify.
Response
According to the result, the neutrophil count decreased the venom control when compared with the normal control. This is consistent with the authors’ discussion and interpretation. The neutrophil count only increased after the treatment with MOE. See Table 2
Comment6- line 430: change anti-venom for anti-ophidic.
Response
Antivenom has been re-written as Anti-ophidic
7- line 445: This is not a better reference for this sentence. Change for: Zuliani et al., 2020.
Response
The suggested reference has been inserted.